# Mixed-Mating Model of Reproduction Revealed in European *Phytophthora cactorum* by ddRADseq and Effector Gene Sequence Data

**DOI:** 10.3390/microorganisms9020345

**Published:** 2021-02-10

**Authors:** Matěj Pánek, Ivana Střížková, Miloslav Zouhar, Tomáš Kudláček, Michal Tomšovský

**Affiliations:** 1Crop Research Institute, Team of Ecology and Diagnostics of Fungal Plant Pathogens, Drnovská 507/73, 161 06 Praha, Czech Republic; iva.strizkova@gmail.com; 2Department of Plant Protection, Faculty of Agrobiology, Food and Natural Resources, Czech University of Life Sciences in Prague, Kamýcká 129, 165 00 Praha, Czech Republic; zouharmiloslav@gmail.com; 3Department of Forest Protection and Wildlife Management, Faculty of Forestry and Wood Technology, Mendel University in Brno, Zemědělská 3, CZ-613 00 Brno, Czech Republic; tomas.kudlacek@mendelu.cz (T.K.); tomsovsk@mendelu.cz (M.T.)

**Keywords:** *Phytophthora cactorum*, effector genes, population structure, reproduction, mixed-mating system

## Abstract

A population study of *Phytophthora cactorum* was performed using ddRADseq sequence variation analysis completed by the analysis of effector genes—RXLR6, RXLR7 and SCR113. The population structure was described by F-statistics, heterozygosity, nucleotide diversity, number of private alleles, number of polymorphic sites, kinship coefficient and structure analysis. The population of *P. cactorum* in Europe seems to be structured into host-associated groups. The isolates from woody hosts are structured into four groups described previously, while isolates from strawberry form another group. The groups are diverse in effector gene composition and the frequency of outbreeding. When populations from strawberry were analysed, both asexual reproduction and occasional outbreeding confirmed by gene flow among distinct populations were detected. Therefore, distinct *P. cactorum* populations differ in the level of heterozygosity. The data support the theory of the mixed-mating model for *P. cactorum*, comprising frequent asexual behaviour and inbreeding alternating with occasional outbreeding. Because *P. cactorum* is not indigenous to Europe, such variability is probably caused by multiple introductions of different lineages from the area of its original distribution, and the different histories of sexual recombination and host adaptation of particular populations.

## 1. Introduction

*Phytophthora cactorum* (Lebert and Cohn) J. Schröt. has a cosmopolitan distribution [1] and the ability to cause infection in more than two hundred plant species [1]. It is considered one of the most important *Phytophthora* pathogens [2], the substantial economic impact of *P. cactorum*, especially on strawberry production, has been repeatedly reported [3,4,5]. The host specificity phenomenon has been verified in various *Phytophthora* species [6,7], as well as in *P. cactorum*, despite its wide host spectrum [1]. Van Der Scheer [8] reported the selective ability of some *P. cactorum* isolates to cause crown rot in strawberry plants. Isolates obtained from other hosts were unable to cause the disease in this crop. Similar conclusions were also revealed by other authors; an association between the host specificity and genetic differences of strawberry crown rot isolates and those from other hosts was discovered [3,4,9,10,11,12]. Recently, four genetic groups were described in *P. cactorum* [13], which also include *P. hedraiandra* [14] and *P. × serendipita* [15,16] previously considered as distinct species. Although the differentiation of genetic lineages in *P. cactorum* was repeatedly confirmed, the genetic background of host specificity remains unclear, although some works indicate genes of virulence as being responsible for host specificity in other *Phytophthora* species [17]. The small cysteine-rich secretory protein (SCR), Crinkler family proteins (CRN) and RXLR genes (genes coding for proteins characterised by the presence of a signalling peptide on the N-terminal region with the following order of amino acids: R-arginine, X-some amino acid, L-leucin, R-arginine) are usually mentioned as being responsible for virulence and host specificity in *Phytophthora* [17,18,19,20].

The occurrence of multiple genetic lineages and the host specificity in *P. cactorum* is not likely to be in agreement with the considered low genetic diversity of a homothalic oomycete [12,21,22]. However, according to some results, the homo- or heterothallic mode of sexual reproduction in *Phytophthora* can be changed. Examples of isolates with homothallic behaviour belonging to heterothallic species have been reported [23,24,25,26]. The concept of homo/heterothallism in *Phytophthora* is considered to be “hormonal homo/heterothallism”, where the sexual behaviour of the isolate depends on the production of one of two sexual hormones [27]. Homothallic species are considered to produce and to be sensitive to both hormones [23,28]. Preferences of particular isolates to produce either antheridia or oogonia in heterothallic species have been recorded; however, bisexuality is hypothesised in homothallic species [29], therefore, the possibility of the mating of two isolates of homothallic species cannot be excluded. Outbreeding was documented in either heterothalic *P. infestans* [30,31,32] or *P. ramorum* [33], or in homothallic species such as *P. sojae* and *P. porri*. In homothallic *Phytophthora* spp., outbreeding was reported as a possible mechanism generating new pathotypes able to overcome host resistance [34,35,36,37]. Outbreeding has a crucial impact on the genetic variability, gene flow and heterozygosity of oomycete populations as well as on their phenotypic features, such as their aggressivity and fitness [38].

*P. cactorum*, as a homothallic species with described host specificity, a wide host spectrum, worldwide distribution and well-documented interspecific hybridisation, is an interesting model for a genotyping study. The aims of this study were to analyse the genetic structure and sequence variation in *P. cactorum* populations from strawberry fields in the Czech Republic and from other hosts, and to evaluate the relationships between these populations, as well as their relationships with the genetic groups defined earlier. We also related the population structure to the host specificity and related this specificity to particular arrangements of some avirulence gene groups. We inferred those conclusions on the basis of SNP (single nucleotide polymorphism) markers generated using high-throughput sequencing with the ddRADseq method of gene library preparation, enabling us to calculate many population characteristics. The ascertained structure was confirmed by phylogeny based on the sequences of three cytoplasmic effector genes.

## 2. Materials and Methods

### 2.1. P. cactorum Isolates

To obtain *P. cactorum* isolates, 24 strawberry plantations in the Czech Republic were sampled. Most of the isolates were isolated from symptomatic strawberry plants in 2017–2019 using the baiting method with subsequent cultivation on a selective V8 PARPNH medium [39]. Except for newly isolated cultures, several DNA samples from isolates used in a previous study [13] were also used to compare genotypes of new isolates with the previously defined genetic groups C1, C2, F and H. Most of these isolates were pathogenic on woody hosts. In total, 136 isolates were used (Table 1). For the purposes of this study, *P. cactorum* isolates originating from the same strawberry plantation were considered a single population and, similarly, each previously defined group (C1, C2, F and H) was considered a separate population. The species identity of the newly acquired isolates was preliminarily morphologically determined according to the shape and size of reproductive organs.

### 2.2. DNA Extraction

All isolates were cultivated on V8 agar plates covered by a cellophane membrane in Petri dishes. After a colony was developed, the mycelium was harvested and homogenised in a 1.5 Eppendorf tube by a mixer mill (Retsch, MM400, Retsch GmbH, Haan, Germany). The extraction of DNA was performed using a DNeasy Power Plant Pro Kit (Qiagen Ltd., Manchester, United Kingdom) according to the manufacturer’s instructions.

### 2.3. Species Identification

Isolates were identified according to DNA barcoding based on the sequences of the ITS region of the ribosomal DNA. The PCR was processed using ITS1 and ITS4 primers designed by White et al. [40], according to a method used by Pánek et al. [13]. The identity of the ITS sequences of *P. cactorum* was confirmed by the BLAST algorithm of the NCBI database.

### 2.4. DNA Library Preparation for Genotyping by ddRADseq

The ddRADseq DNA library was prepared according to the method of Peterson et al. [41]. The composition of the restriction mix of each *P. cactorum* isolate was: 300 ng of sample DNA, 5 U of MspI and 7 U of Sau3AI endonucleases and 1 × Tango buffer (all components from Thermo Fisher Scientific, Inc., Waltham, Massachusetts, USA), added to nuclease-free water to 30 µL. The reaction was performed in a thermocycler at 37 °C for three hours. Single-stranded oligos for the adapter were synthesised by the MilliporeSigma(Burlington, Massachusetts, USA) according to a template proposed by Peterson et al. [41], of which ends were redesigned to meet the MspI and Sau3AI restriction sites’ characteristics. The double-stranded adapters (one for MspI and twelve for Sau3AI) were prepared according to Peterson et al. [41]. The adapters’ ligation onto DNA fragments prepared in a restriction reaction was carried out in a mixture containing 40 ng of fragmented DNA, with both adapter stocks in a final concentration of 0.075 pmol/µL of Sau3AI and 0.041 pmol/µL of MspI adapters, 1 × ligation buffer (Thermo Fisher Scientific, Inc.) and 5 U of T4 DNA ligase (Thermo Fisher Scientific, Inc.) added to nuclease-free water to 40 µL. The reaction conditions were: 23 °C/60 min, 65 °C/10 min, and then the mixture was cooled by 0.6 °C/min until the temperature reached 0 °C. The length selection of DNA fragments in twelve partial DNA libraries was carried out using Pippin prep (Sage Science, Inc., New Castel upon Tyne, United Kingdom), and the selection window was set to 170–370 bp. Where required by protocol, the samples were purified using AMPure XP beads (Beckman Coulter, Brea, California, USA). The DNA concentration in partial libraries was increased by PCR using primers complementary to adapter sequences (one for the Sau3AI and twelve for the MspI adapter). Using the combination of twelve adapters and twelve primers, each sample was characterised by a unique combination of barcodes on both ends of the DNA fragments. The PCR master mix included 20 ng of DNA, 0.5 µM of both primers, 20 µM of dNTP mix, 1 U of Phusion DNA polymerase and 1 × final of Phusion HF buffer (New England BioLabs, Inc., Ipswitch, Massachusetss, USA). The PCR conditions were: 98 °C/45 s, 10 cycles consisting of 98 °C/10 s, 58 °C/10 and 72 °C/15 s and 72 °C/5 min as a final extension. The content of DNA fragments and their length distribution in each partial DNA library was evaluated by Agilent TapeStation (Agilent Technologies, Inc., Santa Clara, California, USA), and the concentration was fluorimetrically measured. The total library was finalised by an equimolar mixture of all twelve partial libraries.

The sequencing on an Illumina MiSeq device was processed by the Biotechnology and Biomedicine Centre of the Academy of Sciences and Charles University (Vestec, Czech Republic) using a MiSeq Reagent Kit v3 (Illumina, Inc., San Diego, CA, USA).

### 2.5. Effector Genes Sequence Analysis

Three effector gene regions that are potentially polymorphic were chosen: RXLR6, RXLR7 [19] and SCR113 [42]. Only a subset of all *P. cactorum* isolates were used in this part of the whole study (Table 1). Primer sequences were used according to Chen et al. [19,42]; their sequences are given in Table 2. The PCR was performed using Phire Plant Direct PCR Mastermix (Thermo Fisher Scientific^TM^). The concentration of primers in RXLR6 and RXLR7 was 0.2 mM, and in SCR113 it was 1.32 mM. The reaction conditions in the thermocycler (Eppendorf Nexus X2, Eppendorf, Hamburg, Germany) were identical for all three DNA regions except for the annealing phase. The cycling conditions were as follows: 98 °C for 5 min; 35 cycles of 98 °C for 30 s, annealing −54 °C for 15 s for both RXLRs and 55 °C for 30 s for SCR113, 72 °C for 60 s, then 72 °C for 5 min. The PCR product was sequenced by MacroGen Inc. (Seoul, South Korea).

### 2.6. Data Processing—ddRADseq

The quality of data in all 2 × 12 individual runs of sequencing on an Illumina MiSeq device was evaluated using FastQC software (Simon Andrews; FastQC version 0.11.8; A quality control tool for high throughput sequence data; Babraham Bioinformatics, Open source, 2010; Cambridge, United Kingdom) [43]; in particular, the total number of sequences, the length of sequences and the per-base quality were ascertained.

The subsequent data processing was carried out using seven modules of Stacks software (Julian Catchen; Stacks version 2.0—analysis tool set for population genomics; Molecular Ecology, 2013) [44,45]. The initial data processing was performed using the “process_radtags” module of the Stacks software. The samples were demultiplexed using combinatorial barcodes, checked for the presence of restriction sites and marked by their original ID numbers. The reads with a low quality score and those with uncalled bases were discarded, along with those which did not contain both restriction sites. All sequences were truncated to a length of 75 bp. The “ustacks” module was used to make short-read sequences into “stacks” (sets of exactly matching sequences). Afterwards, SNPs were identified using the calling by maximum likelihood framework. The settings of this process were: Minimum depth of coverage required to create stacks—3, maximum distance allowed between stacks—2. The catalogue of consensus loci of all samples was created by the “cstacks” module; the number of allowed mismatches between sample loci was set to 1 according to the standard settings of this step. The “sstacks” module searched the loci of all samples created by “ustacks” against a catalogue created by “cstacks”. The data were than transposed by the “tsv2bam” module, to be oriented by locus instead of samples, and in this step sets of paired end reads were also pulled together. The subsequent module “gstacks” removed PCR duplicates, identified SNPs within the whole metapopulation for all loci and phased them into haplotypes. The last module, “populations”, was used to compute statistics on a population level and to create data files for subsequent analyses by another software. For the purpose of evaluating the genetic differentiation among populations, the F-statistics were calculated. Since different methods of calculation emphasise different properties of population data, analogues of the F-statistics were calculated using four distinct methods to obtain more reliable information about population structure: the Φ_st_ statistics were calculated according to Excoffier et al. [46] and, as other analogues of this statistic, the F_st_ according to Meirmans [47], F_st_ according to Weir and Cockerham [48] and D_st_ according to Jost [49] were calculated. The information of genetic differentiation among populations was complemented by the determination of the “number of private alleles” in each group, which also provided information about the mutual isolation between populations. The “frequency of polymorphic loci” and “mean frequency of the most frequent alleles” of each population were used as tools to provide information on the genetic variability of populations. In addition, estimates of “observed” and “expected hetorozygosity” and population “inbreeding coefficient—F_is_” and its variance were calculated. Since the heterozygosity is strongly influenced by the reproduction behaviour of the population, heterozygosity reveals the sexual or asexual nature of the population. Briefly, heterozygosity is increased by clonal reproduction and by outbreeding [50,51], while inbreeding and mitotic recombination cause the reverse [20,52]. Similar information is also traceable by the inbreeding coefficient and its variability, which were also calculated.

To more reliably deduce the structuring of the population of *P. cactorum* and the number of discrete genetic groups (K-numbers), the Structure analysis (Jonathan K. Pritchard; Structure version 2.3.4; Tool for inference population structure using multilocus genotype data, University of Oxford, Oxford, United Kingdom and Pritchrd Lab, Stanford university, Stanford, California, USA) [53] was processed for K = 1 to K = 8. The burning period was set to 50,000 and the total number of MCMC (Markov chain Monte Carlo) repetitions was 500,000 after burning in a no-admixture model without linkage. The structure analysis was repeated ten times for each K number. The results of single runs of structure in terms of the analysis of each K number were compared, and the most probable K number was chosen by the structure harvester (Dent Earl, STRUCTURE HARVESTER, a website and program for visualizing STRUCTURE output and implementing the Evanno method, vA.2, Univesity of California, Santa Cruz, California, USA) [54]. VcfTools (Adam Authon; VCFtools v0.1.13, the program package designed for working with VCF files equipped by suite of functions for use on genetic variation data, Wellcome Trust Sanger Institute, Cambridge, United Kingdom) [55] was used to evaluate the reading depth of each DNA sample. Using the same tool, the individual heterozygosity of each sample was calculated, which provided information on genetic polymorphism on an individual level. The other way of evaluating genetic polymorphism used was to calculate the individual inbreeding coefficient (F_is_), also by VcfTools. The indication of population exposition to some evolutionary pressure was tested by the exact test for Hardy–Weinberg equilibrium [56], calculated by VcfTools as well. The same software was used for the calculation of kinship coefficient (Φ) [57], which expresses the degree of relatedness between isolates. The values of the kinship coefficient (Φ) acquired from this analysis were categorised into five classes according to the following threshold values: (A) 0.5 ≥ Φ ≥ 0.25, (B) 0.25 ≥ Φ ≥ 0.125, (C) 0.125 ≥ Φ ≥ 0.0625, (D) 0.0625 ≥ Φ ≥ 0, (E) Φ ≤ 0 [57]. The threshold values of Φ are matched to relations: 0.5—monozygotic twins, 0.25—parent–offspring/full siblings, 0.125—2nd degree, such as half-sibs, 0.0625—3rd degree, such as first cousins and 0.0—unrelated; negative values indicate rather distinct populations. The relative frequency of samples in each class was ascertained.

### 2.7. Data Processing—Effector Genes (RXLR6, RXLR7 and SCR117)

To reveal the relation between the host specificity, the population structuring and the presence of the specific group of effector genes in different parts of population, the sequence analysis of three of effector genes was conducted. The DNA sequences were manually aligned using the software BioEdit 7.1 (https://bioedit.software.informer.com/7.2/; accessible 9 February 2021). The phylogenetic analysis was performed for all three loci, which were concatenated into one dataset and analysed using maximum likelihood and a Bayesian approach. A partitioning scheme and the best-fit model of evolution were selected in PartitionFinder 2 (Rob Lanfear, PartitionFinder 2—free open source software to select best-fit partitioning schemes and models of molecular evolution for phylogenetic analyses; Australian National University, Canberra, Australia) [58] using the corrected Akaike information criterion, and both linked and unlinked branch lengths were carried out. All available models were chosen by the program and, in total, 84 models were included in the analysis, with all possible partitioning schemes. The selected partitioning schemes were used for both maximum likelihood and Bayesian analysis. Selected evolutionary models were used for the maximum likelihood analysis, while for the Bayesian inference, the averaging model was used. A maximum likelihood phylogeny was constructed by means of RAxML-NG software (Alexey M. Kozlov, RAxML tool for Maximum-likelihood based phylogenetic inference, Heidelberg Institute for Theoretical Studies, Heidelberg, Germany) [59], and a sufficient number of bootstrap replicates was determined by the MRE -based (majority rules method) bootstrapping test [60]. The convergence cutoff value was set to 0.03. Support values were calculated using the transfer bootstrap expectation method [61]. Bayesian inference was carried out in BEAST 2 software (M.A.Suchard, Beast v1.10.4, cross-platform program for Bayesian analysis of molecular sequences using MCMC; University of Auckland, Auckland, New Zealand) [62] using the standard template. A model for each partition was selected automatically via model averaging implemented in the bModelTest package (Remco R. Bouckaert, bModelTest, tool for infering site models in a phylogenetic analysis using MCMC proposals; University of Auckland, Auckland, NZ) [63] using 20,000,000 MCMC repetitions sampled by every 2000th repetition. The posterior parameter estimates were summarised in Tracer (A. Rambaud, Tracer v1.7.1, graphical tool for visualization and diagnostics of MCMC output; University of Edinbourgh, Edinbourgh, United Kingdom and University of Ackland, Auckland, New Zealand) [64]. The minimum ESS value evaluating the quality of posterior estimates was adjusted to 200. The nucleotide diversity [65] was calculated, an AMOVA test was performed and fixation index F_st_ [48] between genetic groups of isolates for all three gene loci was calculated using the software Arlequin 3.5.2.2 (Laurent Excoffier, Arlequin ver 3.5, An Integrated Software Package for Population Genetics Data Analysis, University of Berne, Bern, Switzrerland) [66].

## 3. Results

### 3.1. Analyses Based on ddRADseq Data

Illumina MiSeq sequencing and data processing: The sequencing on an Illumina MiSeq device resulted in a total of 33.6 × 10^6^ reads. The error rate of sequencing was 0.16% in run 1 and 0.23% in run 2. In all reads, the average percentage of GC content was 52.78%, and the average percentage of target sequence length (80 bp) was between 95.3 and 98.4% for all reads. Both the median and lower quartiles of per-base sequence quality were higher than 34, which means a very good quality of reads. During data processing, 7.25% of reads were discarded because of the low barcode quality, 1.34% of reads were of poor sequence quality and in 0.78%, no restriction enzyme cut site was found, thus, 3.05 × 10^7^ (90.64%) reads were retained for further processing. After the processing of all data and SNP calling, the mean number of identified sites per individual was 3943.37 and the average depth of sites per individual was 5.69.

Structure: According to structure analysis, the most probable model appears to be K = 5, with the highest value of ΔK = 677.6, followed by a considerably less probable K = 2 with ΔK = 360.9 (Table A1). Since the more probable K = 5 is also supported by the value of LnP(K), K = 5 was evaluated as the overall more probable choice. In all subsequent analyses, five genetic groups were considered. In this grouping, the strains isolated from strawberry plants form a separate genetic group “S” (strawberry) and include the majority of strawberry populations, except for population 45, which is closer to C1. The C1, C2, H and F groups from woody hosts are well delimited (Figure 1).

Private sites: The numbers of private sites in S group populations were mainly between 0 and 54, which supports the close relations of all populations of the S group, and the numbers of private sites of other groups confirm the dissimilarity of this genetic group from the others. Population 45 was exceptional, with 403 private sites, which places this population outside the S group. The numbers of private sites in the C1, C2, F and H groups were as follows: C2-728, C1-4651, H-3700 and F-8962. These values (Table 3) support the grouping resulting from the structure analysis and indicate the relevance of mutual differentiation between groups well. The presence of larger numbers of private sites is another mark of mutual isolation between groups [67].

Hardy–Weinberg (HW) equilibrium: Hardy–Weinberg equilibrium was confirmed in 14,928 positions of all 24,412 (*p* ≤ 0.05), therefore, 61.2% of loci are in HW equilibrium. Such a result means deviation from HW equilibrium, which indicates the presence of some evolutionary forces influencing the *P. cactorum* population. The deviation from HW equilibrium as such does not explain the type of these forces, but supports conclusions about the presence of some of these forces.

Genetic differentiation between populations (F_st_, Φ_st_, D_st_): The basic structure is similar in all four matrices displaying the differentiation among populations calculated according to various methods. The separation of all groups identified by structure analysis is apparent in the results of F_st_ according to Weir and Cockerham [48] (Table 4). Similar results were reached through the use of other calculation methods—Φ_st_ [46] (Appendix B, Appendix A), F_st_ [47] (Appendix B, Appendix A) and D_ST_ [49] (Appendix B, Appendix A). Since all of those methods provide the measure of population differentiation, the ascertained values of calculated coefficients provide the conclusion that all five groups (S, C1, C2, F and H) are rather separated. The differentiation of the vast majority of populations in the S group is mostly low; population 45 is closer to C1.

P—mean frequency of the most frequent allele in each population: This characteristic is in close association with the homozygosity. In variant positions, the values of the mean frequency of the most frequent alleles in the *P. cactorum* S group are usually between 0.93–0.96, except for population 28 (0.863); the substantially different values of other populations are 0.819 in C2 and 0.744 in F. The ascertained values mean rather high homozygosity in all groups, with mentioned exceptions. The reduced values in those populations means a decrease in their homozygosity in comparison to others, which can be interpreted as being a consequence of increased outbreeding. Detailed information is in Table 3.

Heterozygosity: The heterozygosity values at the population level ranged from 0.060 to 0.465; in all cases, the observed heterozygosity was higher than the expected heterozygosity (Table 3). The highest values were found in groups F (0.465) and C2 (0.355); the lowest value was in C1 (0.060). Differences between groups were also obvious in the heterozygosity values of individual isolates (Appendix A), which increased in groups F, C2 and H; thus, the total heterozygosity of those groups is not the result of a mixture of different genotypes, but more probably the result of an increase in individual heterozygosity in those populations. The individual values in the S group were rather variable (0.060–0.274), which leads to conclusion that the whole group is made up of many genetically different individuals, rather than a few clonal lineages. The ratio between estimates of “observed” and “expected” heterozygosity in all populations indicates an isolation-breaking effect—the mixing of previously isolated populations.

F_is_—inbreeding coefficient: The inbreeding coefficient F_is_ [51] was slightly negative in most populations of the S, F and C2 groups. Since the inbreeding coefficient is defined by the equation F_is_ = 1 − HI/HS, where HI means individual heterozygosity and HS means the subpopulation heterozygosity [68], values lower than zero are determined by the presence of individuals that are more heterozygous than expected given the whole subpopulation, which indicates the presence of some individuals originating from another population. The increasing variance of F_is_ indicates asexual reproduction. The ascertained values thus indicate at least occasional outbreeding; only the C1 (0.001) and H (0.043) groups had slightly positive values. These results suggest that outbreeding occurs with a different frequency in each group, which is also implied by the variability of individual F_is_ values. The values of population and individual F_is_ are summarised in Table 3 and Appendix A.

Nucleotide diversity Π and proportion of polymorphic sites: Nucleotide diversity is a measure of genetic variation, which is similar to expected heterozygosity. The values of nucleotide diversity Π are given in Table 3. Neglecting the values of populations represented by only one sample, the values of S group populations and the C1 group were on the scale of 0.044 to 0.097, while the values of F, C2 and H were 0.451, 0.337 and 0.148. Similarly, the variance of Π (Table 3) had also substantially increased in those three groups. Another measure of genetic polymorphism is the proportion of polymorphic sites, which expresses the percentage of variable loci in a population. This measure is in rough agreement to a previous one: The groups C2 and F also have an increased proportion of polymorphic sites in comparison to other groups. Those results indicate increased outbreeding of the F, C2 and probably also H groups.

Relations between isolates: The kinship coefficient (Φ) values among all isolates, calculated according to Manichaikul et al. [57], are summarised in a relatedness matrix (Appendix A). The C1, C2, H and F groups are rather distant from the S group. The Φ values of either S group and of all groups were divided into five categories A–E, and relative frequencies of kinship coefficient (Φ) values falling into each category were expressed (Table 5). Most values of the S group (54.4%) analysed separately fell into category B (such as parent–offspring/full siblings). If the analysis also included another groups (C1, C2, F and H), most values fell similarly into category B, while a proportion of category E (genetically distinct individuals) markedly increased compared to the S group alone. Only a markedly low proportion (0.9 and 1.2%, respectively) of relations was on the level of clones in the S group and overall. The expected distribution of relatedness in the population under the validity of the presumption of exclusive homothallic inbreeding and clonal reproduction should reveal the presence of only a limited number of lineages comprising relations among isolates of category A. Those lineages should be mutually unrelated (category E), while the numbers of relations in middle categories should be negligible. Compared to that expectation, highly distant relations are notably recorded only if groups other than the S group are included in the analysis. Our results indicate a rather wide scale of ascertained relations in the S group, with the majority of relations falling into middle categories (B and C) and only a significant minority of very close or very distant relations. A possible explanation for this distribution is the presence of outbreeding as a mechanism increasing the proportion of medium-distance relations.

### 3.2. Analyses Based on Sequences of Effector Genes

Phylogeny: The phylogenetic analysis based on DNA sequences of RXLR6, RXLR7 and SCR113 effector genes resulted in a phylogenetic tree (Figure 2). The main statistically supported clades represent groups delimited by structure analysis; however, some isolates were placed in different groups than they were in structure analysis: Isolates 17_45_1a, 17_45_1b (both of them are the only isolates of population 45) and 17_15_1 isolated from strawberry plants clustered together with isolates of the C1 group; on the contrary, isolate 634/13 from C1 clustered together with the S group. Similarly, the clade of C2 also included isolate CBS111725, belonging to the H group, and one separate lineage is composed of two isolates from the H group and one from the F group.

AMOVA: The results of AMOVA based on DNA sequence data of the three examined effector genes are not in complete concordance (Table 6). The resultant F_st_ values congruently put the C2, F and H groups mutually rather close. The locus SCR113 clustered together the S and C1 groups, while the locus RXLR6 clustered the S group together with the C2 group.

Heterozygosity (effector genes): Values of heterozygosity (as nucleotide diversity) calculated for each group based on data for three effector genes are summarised in Table 7. In comparison to the overall heterozygosity calculated from ddRADseq data, these values are substantially higher, except for the SCR113 locus in the C1 group.

### 3.3. Summary of Results—DdRADseq + Effector Genes: Relationship between C1, S and Other Groups

The results of structure based on ddRADseq data as well as phylogeny based on three effector genes congruently support the division of tested isolates into the groups described earlier (C1, C2, F and H) and group S specific to strawberry plants; this result is also supported by the numbers of private alleles (Table 3). The S group is closer to C1 than to others according to effector gene phylogeny (Figure 2), structure (Figure 1) and fixation index F_st_ (Table 4).

The populations of the S group are genetically homogenous, which is confirmed by low fixation index F_st_ or D_st_ values (Table 4, Appendix A) and by kinship coefficient Φ (Appendix A). Despite this proximity, differentiations among populations in the S group were detected, and their differences in genetic variation are obvious (Appendix A and Table 3).

Several isolates that are genetically distinct from members of the S group were obtained from strawberry plants. Isolate 19_28_2 (locality 28) is distant from the members of the S group according to kinship coefficient Φ (Appendix A), and isolate 17_15_1 (locality 15) is closer to C1 than to the S group according to phylogeny based on effector genes; however, according to the kinship coefficient, it is not different from the S group. The separation of isolates 17_15_1 and 19_28_2 from the S group is also not supported by the number of private alleles. Although population 45 includes only two isolates (17_45_1a and 17_45_1b), a relatively large number of private alleles were detected here, which separate this population from both the S and C1 groups. Two isolates from locality 45 appear closer to the C1 group than to the S group according to the fixation index F_st_ value (Table 4). These two isolates are not genetically identical according to the kinship coefficient. Such examples confirm an occasional gene flow between the C1 and S groups.

The C1 group is likely to be genetically homogenous, with low heterozygosity and low probability of outbreeding. On the contrary, the C2 and F groups seem to be highly heterozygotic, with a higher probability of outbreeding in their history. Group H turned out to be highly heterozygotic, but with a low probability of outbreeding documented by inbreeding index F_is_ (Table 3 and Appendix A).

## 4. Discussion

Historically, two host-specific types of *P. cactorum* were distinguished, associated with either strawberry crown rot or woody hosts; this concept has been confirmed and validated in different parts of the world [3,5,8,9,69]. The phenomenon of narrow host specificity of some *Phytophthora* species is in contrast to the extremely wide host spectrum of others, such as *P. cinnamomi* [70] or *P. ramorum* [71]. Intraspecific variability of host specificity, or host preference of some strains, was also repeatedly reported in *P. parasitica* [72,73]. On the contrary, narrow host specificity is typical for other *Phytophthora* spp., e.g., the *P. alni* complex specific to genus *Alnus* [7] or *P. infestans*, which is specific to *Solanaceae* [6,74]. In *P. cactorum*, four distinct genetic groups were delimited recently [13]. The lineage C1 was considered *P. cactorum* sensu stricto, the H lineage was associated with *P. hedraiandra*, C2 is a hybrid between C1 and H (i.e., *P.* × *serendipita*) and the F group includes exclusively Finnish isolates, also considered to be of a probable hybrid origin. Isolates of the F group were considered as specific to birch [4] and they exhibited a high oogonial abortion rate [13]. Although the host specificity of some isolates to strawberry hosts was repeatedly described [3,9,69], the association between population structure and host specificity was not studied in detail. Our results revealed that isolates of *P. cactorum* from strawberry plants do not belong to the C1 group, as expected in [13], but form a separate S group. Those results are supported by structure and by the phylogeny of effector genes, as well as by kinship coefficient, fixation index (F_st_, Φ_st_, D_st_) and the numbers of private sites of each group.

The presence of those genetic groups, including the newly revealed S group, raises a question of the permanency of such a *P. cactorum* population structure, which is primarily determined by the mode of reproduction. Three basic modes of reproduction are known for homothallic *Phytophthora* species: Asexual clonal reproduction driven by the production of zoospores and chlamydospores, homothallic sexual reproduction (selfing) represented by the production of oogonia and antheridia by a single strain [1] and heterothallic sexual reproduction—outbreeding—between two genetically distinct isolates. Mitotic recombination, which is connected to clonal reproduction [75] can also be hypothesised, but this process was rarely observed in *Phytophthora* and only under high concentrations of strong mutagens [76]. Each mode of reproduction is traceable through population heterozygosity, which is influenced by particular reproduction modes in different ways. While clonal and heterothallic sexual reproduction (outbreeding) maintains or increases the population heterozygosity [50,51], selfing and mitotic recombination act in the reverse direction [22,52]. The reproduction mode is also traceable by means of the values of F-statistics; if only selfing occurs in the population, the heterozygosity decreases to close to zero over several generations [77], and the fixation index (F_st_) among populations increases to 1 [22]. Outbreeding decreases both the fixation (F_st_) and inbreeding (F_is_) indices [78]. In heterothallic *P. andina*, inbreeding coefficient F_is_ values between −0.13 and −0.07 were considered to be associated with outbreeding, or an isolation-breaking effect, which is congruent with the low revealed rate of clones in those populations [79], similar conclusions were made by other authors in other species [50,51]. Heterozygosity values revealed by our analyses in *P. cactorum* are rather low. Anyway, in all cases, the observed heterozygosity was slightly higher than expected, which indicates outbreeding and an isolation-breaking effect [52]. The values of our F_is_ are slightly negative in most groups, with the exception of slightly positive values for C1 and H, while the variance of F_is_ is rather low. The revealed proportion of clones (1.2%) and the proportion of genetically distant individuals in populations of the S group are low (Appendix A, Table 5). If outbreeding were absent, the presence of a rather low number of unrelated clonal lineages would be expected in each locality. The genetic diversity of the S group confirms that outbreeding occurs in those populations. The low F_st_ values between S group populations also suggests the migration of *P. cactorum* genotypes among populations. The migration of *P. cactorum* strains among different plantations can be easily explained by the human-mediated transfer of infected plant material [80].

The heterozygosity of *P. cactorum* was repeatedly confirmed as low [12,21,22], thus, some mechanism preventing the total loss of heterozygosity and Muller’s ratchet threat is likely to occur. This phenomenon [81] is described as the gradual accumulation of deleterious mutations in the genome in the absence of sexual recombination, leading sooner or later to probable species extinction due to the loss of species adaptability to environmental changes. The increase in population heterozygosity as a result of outbreeding in one species was confirmed by Carlson et al. [82]. Such an increase in population heterozygosity leads to an increase in the adaptability and total fitness of the population [83]. Our results revealed that heterozygosity is variable among the populations and at the group level as well. In the case of the C2 and F groups, heterozygosity is rather high (0.3562 and 0.4647, Table 3). The differences in genetic variability among types (crown rot and fruit leather rot types) of *P. cactorum* have already been reported [3,4,5]. Such differences in heterozygosity values can be explained by different sexual behaviour in the various populations, confirmed by different F_is_. A similar population structure was described for heterothallic *P. capsici* in China, where both clonally reproducing populations with low genetic diversity and outbreeding populations with high genetic diversity were found [84].

Heterozygosity similar to our results was also revealed in heterothallic *P. cinnamomi* [85], which may indicate fewer differences in the population structure of homothalic and heterothallic *Phytophthora* spp. Outbreeding in homothallic *Phytophthora* species has already been discussed [9,86], but the frequency and mechanism of this phenomenon has not yet been explained in detail. Our results indicate the presence of at least occasional outbreeding in homothallic *P. cactorum*, having a deep impact on population heterozygosity. In the case of increased population heterozygosity caused by outbreeding, albeit infrequent, the rate of heterozygosity remains fixed for long periods by subsequent multiple clonal reproduction by asexual zoospores, although selfing is also present. This mixed-mating model is in agreement with a deviation from Hardy–Weinberg equilibrium, which was confirmed by our results. A similar reproduction behaviour has already been suggested for *P. heveae, P. citrophthora, P. meadii* and *P. megacarya* [22]. The described mixed-mating model could be considered an adaptation protecting the population against the course of Muller’s ratchet [81]. The mixed-mating model resolves this issue without the need for frequent outbreeding. The strategy of a low frequency of sexual reproduction alternating with frequent clonal reproduction is not unusual among microorganisms, such as fungi and oomycetes [83,87], and such behaviour has the potential to decrease the costs of sexual reproduction while maintaining sufficient genetic variability [51]. Such behaviour also allows for phenotypic plasticity sufficient for adaptation to new hosts to be maintained [88]. Therefore, *Phytophthora* spp. seems to be able to behave flexibly either towards clonal reproduction, inbreeding or outbreeding [82,84]. The high heterozygosity value of the F group, associated with frequent oogonial abortion [13], can be explained by a shift in the balance between clonal reproduction and selfing towards clonal reproduction.

Our analyses revealed substantial heterozygosity in effector genes (most of them belong to RXLR effector genes (RXLRs)); its level varies among loci (Table 7). The heterozygosity values found for effector genes are predominantly substantially higher than those based on whole genome sequencing (Table 3). The high heterozygosity of effector genes explains at least part of the calculated overall heterozygosity. Goodwin [22] presumed that the distribution of heterozygosity in one genome is because of the selection for heterozygotes in effector gene loci. Although the number of analysed effector genes is only a small fraction of the presumed hundreds of total RXLRs, or 14 SCRs [17,18,42,89], the presence of their different genotypes in specific groups is rather obvious and strongly correlates with the adaptation of the S group to strawberry plants. The increased speed of evolution in effector genes in comparison to housekeeping genes in *Phytophthora* spp. has been repeatedly confirmed [90,91,92], and our results are another indication that those genes play a crucial role in heterozygosity [93].

Since the allele composition of virulence genes probably determines the host specificity, their change by outbreeding carries the risk of changes of pathogenicity [94] and has a potential to change the host preference or specificity. Another possible important consequence of revealed gene flow between groups is the spreading of resistance to chemical compounds across *P. cactorum* populations. The development of such resistance has been repeatedly evidenced in this pathogen [95,96,97], and its heritability has also been confirmed [34,98]. Therefore, the issue of the outbreeding of *P. cactorum* plays an important role in the spread of this pathogen to new host species and environments; outbreeding is equally important in chemical plant protection as well.

## Figures and Tables

**Figure 1 microorganisms-09-00345-f001:**
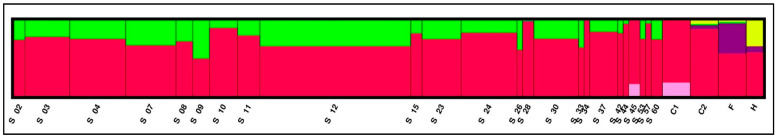
The bar plot of structure results for the most probable division into K = 5 genetic groups. The included populations of isolates from strawberry plants are labelled “S” and by ID of sampled locality; C1, C2, F and H refer to the groups identified earlier [13].

**Figure 2 microorganisms-09-00345-f002:**
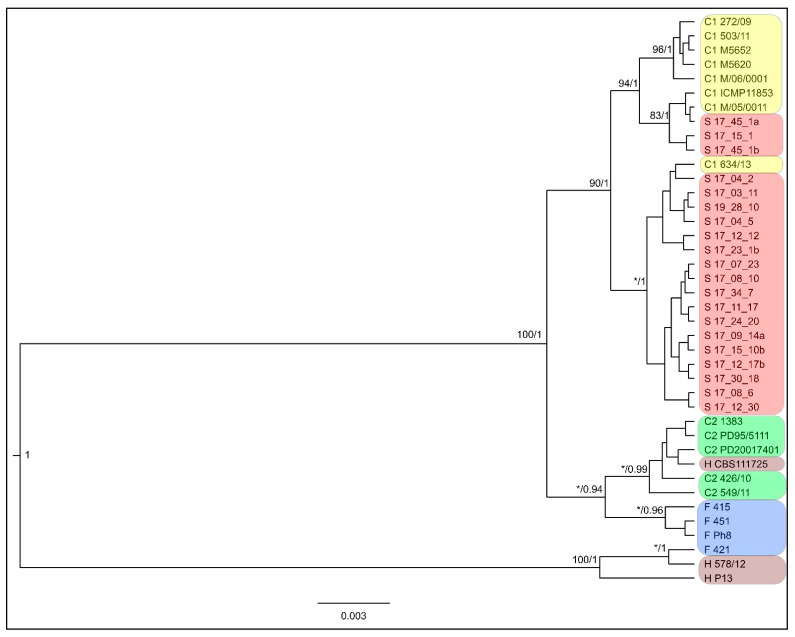
Phylogenetic tree based on three concatenated effector gene loci, RXLR6, RXLR7 and SCR113. The tips of branches are labelled by isolate ID and by the grouping the isolates in genetic groups according to structure analysis results. Numbers at branches indicate maximum likelihood bootstrap proportion and Bayesian posterior probability values. The asterisks (*) mark low support (<75 in maximum likelihood; <90 in Bayesian analysis). The bar indicates the number of expected substitutions per site.

**Table 1 microorganisms-09-00345-t001:** List of isolates used in the study.

Isolate ID	Host Species/Country of Origin	Locality ID	Assignment to Genetic Group	GenBank Accession Number of DNA Sequences		
				RXLR6	RXLR7	SCR113
18_02_3	Fragaria CR	02	S	–	–	–
18_02_1b	Fragaria CR	02	S	–	–	–
17_03_13	Fragaria CR	03	S	–	–	–
17_03_23	Fragaria CR	03	S	–	–	–
17_03_23a	Fragaria CR	03	S	–	–	–
17_03_5	Fragaria CR	03	S	–	–	–
17_03_11	Fragaria CR	03	S	MT896953	MT896979	MT896909
17_03_12	Fragaria CR	03	S	–	–	–
17_03_10	Fragaria CR	03	S	–	–	–
17_03_24	Fragaria CR	03	S	–	–	–
17_04_1a	Fragaria CR	04	S	–	–	–
17_04_9	Fragaria CR	04	S	–	–	–
17_04_8	Fragaria CR	04	S	–	–	–
17_04_12	Fragaria CR	04	S	–	–	–
17_04_2	Fragaria CR	04	S	MT896940	MT896980	MT896910
17_04_10	Fragaria CR	04	S	–	–	–
17_04_5	Fragaria CR	04	S	MT896946	MT896978	MT896900
17_04_7	Fragaria CR	04	S	–	–	–
17_04_7b	Fragaria CR	04	S	–	–	–
17_04_3	Fragaria CR	04	S	–	–	–
18_07_6	Fragaria CR	07	S	–	–	–
17_07_27a	Fragaria CR	07	S	–	–	–
18_07_2S5	Fragaria CR	07	S	–	–	–
17_07_25	Fragaria CR	07	S	–	–	–
18_07_2S1	Fragaria CR	07	S	–	–	–
18_07_14	Fragaria CR	07	S	–	–	–
17_07_12a	Fragaria CR	07	S	–	–	–
17_07_23	Fragaria CR	07	S	MT896941	MT896981	MT896911
17_07_25	Fragaria CR	07	S	–	–	–
17_08_6	Fragaria CR	08	S	MT896948	MT896987	MT896901
17_08_17b	Fragaria CR	08	S	–	–	–
17_08_10	Fragaria CR	08	S	MT896947	MT896988	MT896915
17_09_12	Fragaria CR	09	S	–	–	–
17_09_14a	Fragaria CR	09	S	MT896942	MT896982	MT896921
17_09_14b	Fragaria CR	09	S	–	–	–
18_10_17a	Fragaria CR	10	S	–	–	–
18_10_11	Fragaria CR	10	S	–	–	–
18_10_16	Fragaria CR	10	S	–	–	–
18_10_14a	Fragaria CR	10	S	–	–	–
18_10_18c	Fragaria CR	10	S	–	–	–
17_11_3	Fragaria CR	11	S	–	–	–
17_11_16	Fragaria CR	11	S	–	–	–
17_11_19	Fragaria CR	11	S	–	–	–
17_11_17	Fragaria CR	11	S	MT896950	MT896984	MT896906
17_12_30	Fragaria CR	12	S	MT896954	MT896991	MT896917
17_12_17b	Fragaria CR	12	S	MT896955	MT896993	MT896920
17_12_26	Fragaria CR	12	S	–	–	–
17_12_18b	Fragaria CR	12	S	–	–	–
17_12_5c	Fragaria CR	12	S	–	–	–
17_12_8a	Fragaria CR	12	S	–	–	–
17_12_1b	Fragaria CR	12	S	–	–	–
17_12_7	Fragaria CR	12	S	–	–	–
17_12_18a	Fragaria CR	12	S	–	–	–
17_12_5a	Fragaria CR	12	S	–	–	–
17_12_20	Fragaria CR	12	S	–	–	–
17_12_9	Fragaria CR	12	S	–	–	–
17_12_4	Fragaria CR	12	S	–	–	–
17_12_17a	Fragaria CR	12	S	–	–	–
17_12_28	Fragaria CR	12	S	–	–	–
17_12_24	Fragaria CR	12	S	–	–	–
17_12_31	Fragaria CR	12	S	–	–	–
17_12_16	Fragaria CR	12	S	–	–	–
17_12_25	Fragaria CR	12	S	–	–	–
17_12_3	Fragaria CR	12	S	–	–	–
17_12_6c	Fragaria CR	12	S	–	–	–
17_12_12	Fragaria CR	12	S	MT896951	MT896986	MT896905
17_12_28b	Fragaria CR	12	S	–	–	–
17_12_6a	Fragaria CR	12	S	–	–	–
17_12_6b	Fragaria CR	12	S	–	–	–
17_12_27	Fragaria CR	12	S	–	–	–
17_12_23	Fragaria CR	12	S	–	–	–
17_15_1	Fragaria CR	15	S	MT896963	MT896995	MT896922
17_15_10b	Fragaria CR	15	S	MT896957	MT896992	MT896913
17_23_7	Fragaria CR	23	S	–	–	–
17_23_8	Fragaria CR	23	S	–	–	–
17_23_19	Fragaria CR	23	S	–	–	–
17_23_1d	Fragaria CR	23	S	MT896949	MT896983	MT896908
17_23_3a	Fragaria CR	23	S	–	–	–
17_23_9	Fragaria CR	23	S	–	–	–
17_23_4a	Fragaria CR	23	S	–	–	–
17_24_8a	Fragaria CR	24	S	–	–	–
17_24_4b	Fragaria CR	24	S	–	–	–
17_24_8c	Fragaria CR	24	S	–	–	–
17_24_3a	Fragaria CR	24	S	–	–	–
17_24_26	Fragaria CR	24	S	–	–	–
17_24_12	Fragaria CR	24	S	–	–	–
17_24_20	Fragaria CR	24	S	MT896952	MT896985	MT896903
17_24_5c	Fragaria CR	24	S	–	–	–
17_24_4	Fragaria CR	24	S	–	–	–
17_24_4a	Fragaria CR	24	S	–	–	–
17_26_14	Fragaria CR	26	S	–	–	–
19_28_2	Fragaria CR	28	S	–	–	–
19_28_10	Fragaria CR	28	S	MT896956	MT896989	MT896907
17_30_18	Fragaria CR	30	S	MT896959	MT896994	MT896912
17_30_8	Fragaria CR	30	S	–	–	–
17_30_6	Fragaria CR	30	S	–	–	–
17_30_3	Fragaria CR	30	S	–	–	–
17_30_13	Fragaria CR	30	S	–	–	–
17_30_12b	Fragaria CR	30	S	–	–	–
17_30_12a	Fragaria CR	30	S	–	–	–
17_30_9	Fragaria CR	30	S	–	–	–
18_33_3	Fragaria CR	33	S	–	–	–
17_34_7	Fragaria CR	34	S	MT896958	MT896990	MT896914
17_37_11	Fragaria CR	37	S	–	–	–
17_37_15	Fragaria CR	37	S	–	–	–
17_37_7a	Fragaria CR	37	S	–	–	–
17_37_7c	Fragaria CR	37	S	–	–	–
17_37_13	Fragaria CR	37	S	–	–	–
19_42_2	Fragaria CR	42	S	–	–	–
17_44_12	Fragaria CR	44	S	–	–	–
17_45_1b	Fragaria CR	45	–	MT896966	MT896997	MT896916
17_45_1a	Fragaria CR	45	–	MT896967	MT896998	MT896924
17_53_3	Fragaria CR	53	S	–	–	–
17_57_F1	Fragaria CR	57	S	–	–	–
17_60_25	Fragaria CR	60	S	–	–	–
17_60_26	Fragaria CR	60	S	–	–	–
M5620	Nursery soil CH	–	C1	MT896971	MT896996	MT896918
M/05/0011	Malus BG	–	C1	MT896965	MT897001	MT896898
272/09	Aesculus CR	–	C1	MT896972	MT896999	MT896919
M5652	Nursery soil CH	–	C1	MT896968	MT897003	MT896904
ICMP11853	Malus NZ	–	C1	MT896964	MT897000	MT896897
503/11	Malus CR	–	C1	MT896970	MT897002	MT896902
634/13 *	Malus CR	–	C1	MT896937	MT896977	MT896923
M/06/0001	Fragaria BG	–	C1	MT896969	MT897016	MT896899
426/10	Tilia CR	–	C2	MT896943	MT897013	MT896928
PD95/5111	Idesia NL	–	C2	MT896939	MT897009	MT896931
1383	Arbutus E	–	C2	MT896938	MT897008	MT896926
PD20017401	Penstemon NL	–	C2	MT896945	MT897006	MT896930
549/11	Rhododendron CR	–	C2	MT896944	MT897014	MT896929
421	water FL	–	F	MT896975	MT897012	MT896927
Ph8	Betula FL	–	F	MT896962	MT897010	MT896925
415	Betula FL	–	F	MT896961	MT897011	MT896933
451	Sorbus FL	–	F	MT896960	MT897015	MT896932
CBS111725	Viburnum NL	–	H	MT896973	MT897007	MT896934
P13	Quercus SK	–	H	MT896974	MT897005	MT896936
578/12	Fragaria CR	–	H	MT896976	MT897004	MT896935

Isolates are assigned to genetic groups according to results of structure analysis. An asterisk (*) marks the isolate previously designed as a *P. cactorum* epitype. The “dash” signify that given isolate was not used in particular analysis, the assignment to groups, or locality IDs was not performed as redundant. Acronyms of countries of isolate origin: BG—Bulgaria, CH—Switzerland, CR—the Czech Republic, E—Spain, FL—Finland, NL—the Netherlands, NZ—New Zealand, SK—Slovakia.

**Table 2 microorganisms-09-00345-t002:** List of primers used in amplification of effector genes DNA sequences.

Amplified DNA Region	Primer Name	Primer Sequence 5′ to 3′	Published By
**RXLR6**	PcRXLR6snpF	TCTTCTGAGCCCCCAGTATC	Chen et al., 2014 [19]
	PcRXLR6snpR	CAGGAACACTCCTTGCCTGT	Chen et al., 2014 [19]
**RXLR7**	PcRXLR7snpF	GGGCACTCACATTTCCATCT	Chen et al., 2014 [19]
	PcRXLR7snpR	GACTGCTTCGAGTGTCACCA	Chen et al., 2014 [19]
**SCR113**	16448F_F	ATGAATCCGTCTTTTGAAG	Chen et al., 2018 [42]
	16448F_R	TCATGACTTCCTGGATGAAT	Chen et al., 2018 [42]

**Table 3 microorganisms-09-00345-t003:** Population characteristics calculated on the basis of analysis of ddRADseq sequence data. The displayed characteristics are calculated for each population of the S group and for groups C1, C2, F and H.

Population ID	Genetic Group	Number of Private Alleles	Proportion (%) of Polymorphic Loci	Mean Frequency of th Most Frequent Allele (P)			Heteroygosity (H)						An Estimate of Nucleotide Diversity (Π)			Population Inbreeding Coefficient (Fis)		
				P	Variance	Standard Error	Mean Observed H	Variance	Standard Error	Mean Expected H	Variance	Standard Error	Π	Variance	Standard Error	Fis	Variance	Standard Error
**2**	**S**	**7**	**0.0255**	**0.9585**	0.0188	0.0016	**0.0817**	0.0739	0.0032	**0.0421**	0.0191	0.0016	**0.0776**	0.0668	0.0030	**−0.0061**	0.0051	0.0041
**3**	**S**	**49**	**0.0199**	**0.9642**	0.0153	0.0010	**0.0671**	0.0580	0.0020	**0.0384**	0.0167	0.0011	**0.0568**	0.0400	0.0016	**−0.0171**	0.0190	0.0108
**4**	**S**	**27**	**0.0196**	**0.9685**	0.0138	0.0009	**0.0614**	0.0540	0.0017	**0.0334**	0.0149	0.0009	**0.0489**	0.0345	0.0014	**−0.0207**	0.0157	0.0100
**7**	**S**	**37**	**0.0165**	**0.9679**	0.0139	0.0008	**0.0611**	0.0534	0.0017	**0.0343**	0.0151	0.0009	**0.0484**	0.0327	0.0013	**−0.0211**	0.0191	0.0095
**8**	**S**	**9**	**0.0232**	**0.9612**	0.0174	0.0013	**0.0768**	0.0691	0.0026	**0.0397**	0.0180	0.0013	**0.0688**	0.0566	0.0024	**−0.0123**	0.0077	0.0052
**9**	**S**	**3**	**0.0188**	**0.9681**	0.0145	0.0010	**0.0633**	0.0575	0.0020	**0.0328**	0.0150	0.0010	**0.0540**	0.0429	0.0017	**−0.0146**	0.0085	0.0053
**10**	**S**	**9**	**0.0259**	**0.9614**	0.0172	0.0016	**0.0755**	0.0673	0.0031	**0.0399**	0.0179	0.0016	**0.0671**	0.0540	0.0028	**−0.0138**	0.0110	0.0105
**11**	**S**	**4**	**0.0216**	**0.9634**	0.0165	0.0013	**0.0722**	0.0649	0.0025	**0.0377**	0.0170	0.0013	**0.0661**	0.0552	0.0023	**−0.0096**	0.0071	0.0060
**12**	**S**	**54**	**0.0193**	**0.9667**	0.0137	0.0008	**0.0630**	0.0524	0.0016	**0.0370**	0.0157	0.0009	**0.0439**	0.0236	0.0011	**−0.0341**	0.0336	0.0284
**15**	**S**	**0**	**0.0139**	**0.9621**	0.0175	0.0018	**0.0754**	0.0697	0.0036	**0.0379**	0.0175	0.0018	**0.0754**	0.0694	0.0036	**0.0000**	0.0006	0.0016
**23**	**S**	**7**	**0.0171**	**0.9665**	0.0149	0.0010	**0.0654**	0.0583	0.0019	**0.0350**	0.0157	0.0010	**0.0551**	0.0421	0.0017	**−0.0169**	0.0130	0.0088
**24**	**S**	**35**	**0.0226**	**0.9653**	0.0151	0.0010	**0.0658**	0.0575	0.0020	**0.0369**	0.0162	0.0011	**0.0555**	0.0400	0.0016	**−0.0171**	0.0174	0.0116
**26**	**S**	**2**	**0.0218**	**0.9645**	0.0165	0.0017	**0.0711**	0.0661	0.0033	**0.0356**	0.0165	0.0017	**0.0711**	0.0661	0.0033	**0.0000**	0.0000	0.0000
**28**	**S**	**9**	**0.0086**	**0.8628**	0.0498	0.0075	**0.2745**	0.1994	0.0150	**0.1372**	0.0498	0.0075	**0.2745**	0.1994	0.0150	**0.0000**	0.0000	0.0000
**30**	**S**	**31**	**0.0205**	**0.9688**	0.0137	0.0010	**0.0600**	0.0531	0.0019	**0.0331**	0.0148	0.0010	**0.0493**	0.0356	0.0015	**−0.0176**	0.0155	0.0101
**33**	**S**	**1**	**0.0205**	**0.9664**	0.0157	0.0017	**0.0672**	0.0627	0.0034	**0.0336**	0.0157	0.0017	**0.0672**	0.0627	0.0034	**0.0000**	0.0000	0.0000
**34**	**S**	**0**	**0.0519**	**0.9377**	0.0274	0.0087	**0.1247**	0.1094	0.0174	**0.0623**	0.0274	0.0087	**0.1247**	0.1094	0.0174	**0.0000**	0.0000	0.0000
**37**	**S**	**6**	**0.0236**	**0.9639**	0.0159	0.0013	**0.0709**	0.0627	0.0026	**0.0378**	0.0169	0.0014	**0.0622**	0.0490	0.0023	**−0.0142**	0.0104	0.0087
**42**	**S**	**2**	**0.0293**	**0.9517**	0.0218	0.0027	**0.0965**	0.0872	0.0054	**0.0483**	0.0218	0.0027	**0.0965**	0.0872	0.0054	**0.0000**	0.0000	0.0000
**44**	**S**	**2**	**0.0360**	**0.9455**	0.0243	0.0046	**0.1091**	0.0973	0.0092	**0.0545**	0.0243	0.0046	**0.1091**	0.0973	0.0092	**0.0000**	0.0000	0.0000
**45**	**/**	**403**	**0.0226**	**0.9622**	0.0170	0.0021	**0.0751**	0.0674	0.0042	**0.0388**	0.0175	0.0021	**0.0710**	0.0609	0.0040	**−0.0061**	0.0038	0.0052
**53**	**S**	**2**	**0.0250**	**0.9615**	0.0178	0.0021	**0.0771**	0.0712	0.0042	**0.0385**	0.0178	0.0021	**0.0771**	0.0712	0.0042	**0.0000**	0.0000	0.0000
**57**	**S**	**0**	**0.0403**	**0.9415**	0.0259	0.0050	**0.1171**	0.1035	0.0101	**0.0585**	0.0259	0.0050	**0.1171**	0.1035	0.0101	**0.0000**	0.0000	0.0000
**60**	**S**	**3**	**0.0253**	**0.9599**	0.0180	0.0016	**0.0791**	0.0713	0.0032	**0.0409**	0.0185	0.0016	**0.0745**	0.0636	0.0030	**−0.0069**	0.0049	0.0043
**C1**		**1320**	**0.0261**	**0.9649**	0.0154	0.0013	**0.0604**	0.0535	0.0025	**0.0370**	0.0164	0.0014	**0.0614**	0.0486	0.0024	**0.0014**	0.0156	0.0068
**C2**		**728**	**0.1253**	**0.8185**	0.0548	0.0030	**0.3562**	0.2179	0.0059	**0.1875**	0.0564	0.0030	**0.3366**	0.1955	0.0056	**−0.0309**	0.0278	0.0107
**F**		**8962**	**0.1649**	**0.7439**	0.0511	0.0016	**0.4647**	0.2052	0.0032	**0.2788**	0.0532	0.0016	**0.4511**	0.1654	0.0029	**−0.0206**	0.0861	0.0057
**H**		**3700**	**0.0571**	**0.9165**	0.0312	0.0017	**0.1205**	0.0924	0.0030	**0.0907**	0.0344	0.0018	**0.1483**	0.0999	0.0031	**0.0428**	0.0482	0.0051

**Table 4 microorganisms-09-00345-t004:** The values of fixation index (F_st_) calculated according to Weir and Cockerham (1984).

Population ID	Population ID																												Population ID
	12	60	24	7	37	11	15	3	8	C1	F	4	30	53	44	10	23	H	2	33	28	45	C2	42	34	9	57	26	
**12**		0.04	0.07	0.04	0.06	0.04	0.03	0.07	0.04	**0.68**	0.47	0.04	0.09	0.03	0.03	0.05	0.03	**0.84**	0.04	0.03	0.12	0.61	**0.49**	0.06	0.09	0.03	0.03	0.03	**12**
**60**			0.08	0.08	0.10	0.09	0.11	0.09	0.09	**0.60**	0.34	0.08	0.10	0.10	0.10	0.08	0.08	**0.80**	0.11	0.11	0.19	0.54	**0.38**	0.11	0.16	0.10	0.08	0.08	**60**
**24**				0.09	0.08	0.09	0.08	0.09	0.09	**0.66**	0.38	0.11	0.09	0.07	0.07	0.09	0.09	**0.81**	0.08	0.09	0.09	0.60	**0.43**	0.06	0.10	0.09	0.05	0.09	**24**
**7**					0.09	0.06	0.06	0.10	0.07	**0.70**	0.42	0.06	0.12	0.06	0.07	0.08	0.06	**0.84**	0.07	0.06	0.15	0.65	**0.46**	0.09	0.11	0.06	0.04	0.06	**7**
**37**						0.09	0.11	0.09	0.09	**0.62**	0.35	0.10	0.10	0.09	0.08	0.10	0.10	**0.80**	0.10	0.11	0.08	0.57	**0.40**	0.08	0.13	0.10	0.08	0.10	**37**
**11**							0.08	0.08	0.08	**0.64**	0.35	0.07	0.10	0.09	0.10	0.09	0.07	**0.81**	0.09	0.08	0.09	0.60	**0.40**	0.11	0.08	0.07	0.07	0.08	**11**
**15**								0.08	0.08	**0.59**	0.33	0.07	0.11	0.11	0.11	0.10	0.07	**0.77**	0.09	0.07	0.10	0.58	**0.37**	0.13	0.18	0.08	0.06	0.08	**15**
**3**									0.09	**0.66**	0.38	0.09	0.11	0.08	0.07	0.09	0.10	**0.81**	0.08	0.09	0.13	0.60	**0.43**	0.08	0.11	0.10	0.07	0.09	**3**
**8**										**0.64**	0.36	0.07	0.12	0.08	0.11	0.11	0.08	**0.81**	0.09	0.09	0.15	0.60	**0.41**	0.11	0.13	0.08	0.07	0.08	**8**
**C1**											**0.39**	**0.70**	**0.68**	**0.59**	**0.39**	**0.59**	**0.68**	**0.79**	**0.63**	**0.63**	**0.40**	**0.34**	**0.46**	**0.52**	**0.29**	**0.71**	**0.39**	**0.60**	**C1**
**F**												**0.40**	**0.38**	**0.32**	**0.31**	**0.34**	**0.38**	**0.48**	**0.34**	**0.33**	**0.28**	**0.38**	**0.29**	**0.32**	**0.24**	**0.38**	**0.30**	**0.33**	**F**
**4**													0.12	0.07	0.07	0.09	0.06	**0.83**	0.07	0.08	0.12	0.64	**0.45**	0.11	0.12	0.06	0.06	0.06	**4**
**30**														0.10	0.10	0.10	0.12	**0.82**	0.10	0.11	0.12	0.62	**0.43**	0.07	0.09	0.13	0.07	0.13	**30**
**53**															0.08	0.08	0.08	**0.79**	0.08	0.11	0.10	0.54	**0.36**	0.11	0.12	0.09	0.09	0.07	**53**
**44**																0.08	0.09	**0.70**	0.11	0.12	0.18	0.36	**0.30**	0.07	0.08	0.11	0.10	0.08	**44**
**10**																	0.08	**0.78**	0.09	0.11	0.10	0.55	**0.39**	0.08	0.13	0.10	0.06	0.10	**10**
**23**																		**0.82**	0.07	0.07	0.16	0.63	**0.43**	0.11	0.14	0.06	0.06	0.07	**23**
**H**																			**0.79**	**0.79**	**0.64**	**0.76**	**0.44**	**0.76**	**0.56**	**0.82**	**0.69**	**0.79**	**H**
**2**																				0.10	0.09	0.58	**0.38**	0.12	0.19	0.08	0.08	0.08	**2**
**33**																					0.12	0.58	**0.39**	0.15	0.20	0.07	0.09	0.10	**33**
**28**																						0.39	**0.26**	0.07	0.08	0.20	0.09	0.17	**28**
**45**																							**0.43**	0.47	0.37	0.66	0.39	0.57	**45**
**C2**																								**0.35**	**0.22**	**0.44**	**0.29**	**0.37**	**C2**
**42**																									0.11	0.12	0.09	0.13	**42**
**34**																										0.22	0.06	0.18	**34**
**9**																											0.10	0.07	**9**
**57**																												0.11	**57**

Populations of the S group are marked by their ID number, and the matrix also includes the genetic groups originating from woody hosts—C1, C2, F and H.

**Table 5 microorganisms-09-00345-t005:** Relative frequencies of relationship values between *P. cactorum* isolates categorised into five classes. The threshold values of Φ are matched to relations: 0.5—monozygotic twin or clone, 0.25—parent–offspring/full sibs, 0.125—2nd degree, such as half-sibs, 0.0625—3rd degree, such as first cousins, 0.0—unrelated, negative values indicate distinct populations.

Category	Range of Kinship Coefficient (Φ)	Relative Frequencies of Kinship Coefficient (Φ) Values Falling into Categories A–E	
		Among Samples of All Groups	Among Samples of S Group
A	0.5 ≥ Φ ≥ 0.25	0.9	1.2
B	0.25 ≥ Φ ≥ 0.125	40.0	54.4
C	0.125 ≥ Φ ≥ 0.0625	24.5	33.2
D	0.0625 ≥ Φ ≥ 0	7.5	7.6
E	Φ ≤ 0	27.1	3.6

**Table 6 microorganisms-09-00345-t006:** The values of fixation index (F_st_) calculated for three analysed effector genes RXLR6, RXLR7 and SCR113 for isolate groups C1, C2, F and H originating from woody hosts and the S group from strawberry plants.

**RxLR6**						
F_st_ total		C1	C2	F	H	S
0.38	C1	0.00				
	C2	0.33	0.00			
	F	0.27	0.45	0.00		
	H	0.19	0.44	0.21	0.00	
	S	0.42	−0.04	0.55	0.55	0.00
**RxLR7**						
F_st_ total		C1	C2	F	H	S
0.16	C1	0.00				
	C2	0.14	0.00			
	F	0.15	0.00	0.00		
	H	0.16	0.00	0.00	0.00	
	S	0.20	0.16	0.17	0.18	0.00
**SCR113**						
F_st_ total		C1	C2	F	H	S
0.66	C1	0.00				
	C2	0.69	0.00			
	F	0.72	0.18	0.00		
	H	0.73	0.12	0.09	0.00	
	S	0.00	0.83	0.86	0.87	0.00

**Table 7 microorganisms-09-00345-t007:** Heterozygosity calculated for each genetic group of isolates of *P. cactorum* based on sequence data of three effector genes RXLR6, RXLR7 and SCR113.

Genetic Group	Gene Locus					
	RXLR6		RXLR7		SCR113	
	Gene Diversity	Variance	Gene Diversity	Variance	Gene Diversity	Variance
C1	0.7500	0.1391	0.7500	0.1319	0.0000	0.0000
C2	0.4000	0.2373	1.0000	0.1265	0.8000	0.1640
F	0.7000	0.2184	1.0000	0.1768	0.8333	0.2224
H	1.0000	0.5000	1.0000	0.2722	1.0000	0.2722
S	0.3526	0.1227	0.0016	0.0012	0.0000	0.0000

## Data Availability

The sequence data of three tested effector genes are available in NCBI GenBank under their ID numbers (Table 1). The ddRADseq data are available on request from the corresponding author.

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
