# Peer review of "Mixed-Mating Model of Reproduction Revealed in European Phytophthora cactorum by ddRADseq and Effector Gene Sequence Data"

_microorganisms, 2021, doi:10.3390/microorganisms9020345_

Round 1

Reviewer 1 Report

The paper describes the genetic structure based on polymorphism of SNPs data and three genes, i.e. SCR113, RXLR6, and RXLR7, which are supposed to evolve in a way of adaptation of Phytophthora cactorum to different hosts in Europe. Understanding the molecular structure of these genes, known as effector-genes, became widely accepted as essential for an understanding of the processes underlying the pathogen diversity and appurtenance to the different genetic lineages. The aim of the study was fulfilled because the genetic structure and sequence variation in pathogen populations from strawberry and from some other plant-hosts revealed the difference of P. cactorum from the other oomycete clades. The novelty of this investigation resides in discovering the outbreeding process in the P. cactorum isolates from Fragaria (low Fst values, low and negative Fis coefficient).

About presentation:

  1. 7, lines 183-201: the description of the statistical methods applied for population structure maybe is too long and full of many details but it can provide knowledge to the readers less experienced in population genetics. Otherwise, the length of the paper is commensurate with its content.

Quality of figures and could be improved. I would only propose to add Table 3 as a supplementary one. The figure of partition would better illustrate the Structure clustering of the studied genetic lineages.

Table 4 could be better formatted, without single letters left in the headline. All tables from Appendix should be moved to the Supplementary material.

References are complete and adequate.

About Scientific evaluation:

The general scientific approach is properly stated and well explained. I have no particular comments, the article is well-written, and presents good technical quality. The applied methodology based on SNPs data achieved by ddRAPseq method is appropriate. Some parts of these results (p.13, lines 53 – 63) should be moved to the discussion - fully justified. The analyses based on effector-genes sequences helped to distinguish the inner characteristics of the studied isolates from group S, and together with the SNPs data – to create the distribution of the similarity between all investigated isolates of P. cactorum.

I recommend publishing the article after minor corrections.

Author Response

Dear Reviewer,

On behalf of authors team I would like to express our appreciation of your work and valuable recomendations to our text. According to your suggestions, the description of dataprocessing of three effector genes was somewhat shortened, although some lenght of this part of text was necessarry to maintain for maintaining of its intelligibility. The quality of phylogenetic tree was improved and the Structure bar plot was inserted for better illustration of relations between analysed genetic groups of P. cactorum based on ddRADseq. The tables were organised and formating of some of them was performed according to your suggestion. However, although we considered severely your suggestion to move the part of results (p.13, lines 53 – 63) into discussion, we decided to leave it in original location, since in discussion section those thoughts are examined in wider context (p. 18, line 178 - 189), while in the results section we make effort to explain the importance of each result in detail and separately from others.

Sincerely

Matěj Pánek, corresponding author

Reviewer 2 Report

Dear Authors

Generally I like the paper entitled “Phytophthora cactorum by ddRADseq and effector gene se-3quence data”. The publication is interesting, because the authors analyze  the model of reproduction. The ddRADseq data is used to analyze the structure of the data structure (although they could have chosen the simpler version). Why else did they analyze effector genes? It would be much more useful to analyze the genes that determine the reproductive model. Maybe you could address this issue in the discussion? I also have some other suggestions:

L37-”…despite of its wide host spectrum.” This sentence could be justified with citation e.g. as follows:

Orlikowski, L. B., Duda, B., & Oszako, T. (2004). The occurrence of Phytophthora cactorum on rowan (Sorbus aucuparia) in forest nurseries. Sylwan, 10, 67-72.

114 – “at 37° C” write  37 °C

L123-124 – “23° C / 60min, 65° C / 10 min, and then the mixture was cooled by 0.6° C / min until the temperature reached 0° C.” add space after digit and delete before C

L134 – “98° C/45 s, 10 cycles consisting of 98° C/10 s, 58° C/10 and 72° C/15 s, and 72° C/5…” let the “°” stand nearby C

L151 – “150RXLRs, and 55° C for 30 s for SCR113, 72° C for 60 s; then 72° C for 5 min.” let the “°” stand nearby C like in L150. Please check through the whole manuscript.

L132- “this concept has been confirmed and validated in different parts of the world” I suggest to cite e.g. the below paper:

Nowakowska, J. A., Stocki, M., Stocka, N., Ślusarski, S., Tkaczyk, M., Caetano, J. M., ... & Oszako, T. (2020). Interactions between Phytophthora cactorum, Armillaria gallica and Betula pendula Roth. Seedlings Subjected to Defoliation. Forests, 11(10), 1107.

Author Response

Dear Reviewer,

On behalf of authors team I would like to express our appreciation of your work and valuable recomendations to our text. We accepted all of your suggestions improving some parts of the text. I would also like to explain our intention in selection of gene loci for the second part of analyses. The original intention of our study was to evaluate the relation between the population structure and the host specificity. The main purpose of calculating the values of multiple parameters used in our analyses was thus to infer the population structure and to correlate it to host plant on the rough distinguishing level – woody species/strawberries. The genes determinig virulence seems to be convenient for those purposes, because their products participate directly on infection process and thus their changes has potential to affect directly this process, which was the reason of their selection. I agree with your idea to use directly the genes determinig reproductive behaviour of Phytophthora, but this is only the question of future. To our knowledge, the recognising the genes determining the reproductive behaviour of Phytophthora species is on the very begining. Only last year the 18 candidate genes related to production of sexual hormones in Phytophthora infestans were suggested.

Thank you for suggestions of improving the text by citations. We selected some citations which meet more preciselly main meaning of concerned two sentences in our text.

Sincerely

Matěj Pánek, corresponding author